# PRP4 Induces Epithelial–Mesenchymal Transition and Drug Resistance in Colon Cancer Cells via Activation of p53

**DOI:** 10.3390/ijms23063092

**Published:** 2022-03-13

**Authors:** Salman Ul Islam, Muhammad Bilal Ahmed, Jong-Kyung Sonn, Eun-Jung Jin, Young-Sup Lee

**Affiliations:** 1Department of Pharmacy, Cecos University, Hayatabad, Peshawar 25000, Pakistan; dr_ssulman@yahoo.com; 2School of Life Sciences, College of Natural Sciences, Kyungpook National University, Daegu 41566, Korea; mbilalknu@gmail.com; 3Department of Biology, College of Natural Sciences, Kyungpook National University, Daegu 41566, Korea; sonnjk@knu.ac.kr; 4Department of Biological Science, College of Natural Science, Wonkwang University, Iksan 54538, Korea

**Keywords:** PRP4, miR-210, HCT116, p53, EMT, drug resistance

## Abstract

Pre-mRNA processing factor 4B (PRP4) promotes pre-mRNA splicing and signal transduction. Recent studies have shown that PRP4 modulates the assembly of actin cytoskeleton in cancer cells and induces epithelial–mesenchymal transition (EMT) and drug resistance. PRP4 displays kinase domain-like cyclin-dependent kinases and mitogen-activated protein kinases, making it capable of phosphorylating p53 and other target proteins. In the current study, we report that PRP4 induces drug resistance and EMT via direct binding to the p53 protein, inducing its phosphorylation. Moreover, PRP4 overexpression activates the transcription of miR-210 in a hypoxia-inducible factor 1α (HIF-1α)-dependent manner, which activates p53. The involvement of miR-210 in the activation of p53 was confirmed by utilizing si-miR210. si-miR210 blocked the PRP4-activated cell survival pathways and reversed the PRP4-induced EMT phenotype. Moreover, we used deferoxamine as a hypoxia-mimetic agent, and si-HIF to silence HIF-1α. This procedure demonstrated that PRP4-induced EMT and drug resistance emerged in response to consecutive activation of HIF-1α, miR-210, and p53 by PRP4 overexpression. Collectively, our findings suggest that the PRP4 contributes to EMT and drug resistance induction via direct interactions with p53 and actions that promote upregulation of HIF-1α and miR-210. We conclude that PRP4 is an essential factor promoting cancer development and progression. Specific PRP4 inhibition could benefit patients with colon cancer.

## 1. Introduction

The cell cycle regulatory protein p53 (TP53) is a well-known tumor-suppressor protein. Disruption of the p53-mediated functions can lead to cancer initiation and/or progression. Interestingly, approximately one-half of all human cancers have been associated with mutations in the p53 gene and loss of its function has been observed in over 80% of tumors [1,2,3]. Overexpression of the mutant p53 protein with reduced or abolished function induces resistance to standard antineoplastic therapeutics, including doxorubicin, cetuximab, cisplatin, temozolomide, gemcitabine, and tamoxifen, with the mutations in the p53 gene affecting protein conformation [4,5,6,7,8,9,10]. Additionally, the p53 protein has been implicated in the epithelial–mesenchymal transition (EMT), EMT-associated generation of stem cell properties, and tumor metastasis [11,12].

MicroRNAs (miRNAs) are small, single-stranded, evolutionarily conserved noncoding RNAs, typically 20–24 bp in length, and are expressed in all mammalian cells [13]. MicroRNAs negatively regulate posttranscriptional events by binding to the 3′-untranslated region (3′-UTR) of target mRNA at specific sequences. Such a feature results in silencing or degradation of target mRNA [13,14,15]. A single miRNA can regulate multiple target genes, and a single gene can be regulated by multiple miRNAs [13]. MicroRNAs are estimated to target over 60% of protein-coding genes in the human genome and are involved in diverse physiological processes (i.e., biochemical pathways regulation, signal transduction, cell differentiation and proliferation, and cell death). MicroRNAs also contribute to disease-associated pathophysiology, notably concerning cancer [15,16]. Recent investigations have revealed that miRNAs regulate the expression and function of p53. More than 20 unique miRNAs (i.e., miR-125b, miR-504, miR-33, miR-380-5p, miR-25, and miR-30d) have been identified as direct negative regulators of p53 [17,18,19,20,21,22,23]. By contrast, several miRNAs indirectly promote the activation of p53 by targeting its transcripts that encode regulatory proteins. For example, miR-192/194/215, miR-143/145, miR-29b, miR-605, miR-25, miR-32, miR-18b, and miR-339-5p have been shown to be capable of direct MDM2 repression, indirectly activating the p53 [17,18,24,25]. Likewise, miR-34a has been reported to enhance transcription and encoded protein levels of p53 by targeting its multiple negative regulators (i.e., MDM4, SIRT1, HDAC1, and YY1) [26].

PRP4, a 1007 amino acid protein containing an N-terminal 340-amino acid Arg/Ser-rich domain, has been implicated in pre-mRNA splicing and cell signaling [27,28]. Additionally, PRP4 contains a kinase domain (amino acids 686–1003) sharing homology with cyclin-dependent kinases and mitogen-activated protein kinases [29,30]. We previously showed that PRP4 promoted drug resistance in cancer cell lines as it could induce changes in the cell cytoskeletal architecture and induce EMT [31,32]. Here, we analyzed the interaction between PRP4 and p53 in various cancer cell lines. Additionally, we analyzed PRP4-mediated miRNA’s expression regulation. Among the miRNAs activated by PRP4, we focused on miR-210, as several previous studies have shown its effects in cell proliferation processes [33,34,35]. We found that activation of the PRP4-mediated miR-210s acted on p53, a potentially critical EMT and drug resistance mediator. Our findings demonstrate a novel molecular mechanism that delineates PRP4-induced EMT and drug resistance in colon cancer cells.

## 2. Results

### 2.1. Interaction between PRP4 and p53 Proteins in Various Cancer Cell Lines

We analyzed the transfection effect of PRP4 on the expression of p53 in HT29, Colo 320, PC-3, HCT116, and HCT-15 cell lines. Among these cell lines, the wild-type (WT) p53 protein is present only in HCT116 and HCT-15 cells. The other cells express mutant versions of the p53 protein ([HT29: G→A mutation in codon 273; Arg→His substitution] [Colo 320: position 248, R→W; Arg→Trp] [PC-3: Position 138, GCC {Ala} is deleted]) [36,37]. Overexpression of PRP4 increased the level of p53 in Colo 320, HCT116, and HCT-15 cell lines (Figure 1A (left and right panels) and Appendix A). Still, the HT29 cells showed decreased p53 expression (Figure 1A (left panel)). p53 did not appear in the PC-3 cell lines, even though the mRNA in p53 was visible in the PC-3 cells (Figure 1A (left panel) and Appendix A). We performed siRNA-mediated knockdown of PRP4 using a pool including three target-specific 19–25 nucleotide-long siRNAs (si-PRP4) to determine the correlations between PRP4 and p53. The si-PRP4 inhibits both the p53’s PRP4 and the PRP4-induced expression (Figure 1A (right panel) and Appendix A). We performed an immunoprecipitation targeting total cell extracts from parental and PRP4-transfected cells that allowed us to identify a stable association between PRP4 and the p53 in HCT 116 and HCT-15 cells. However, in PC-3, HT29, and Colo 320 cells, the PRP4 and p53 did not coimmunoprecipitate (Figure 1B). Next, we analyzed the kinase activity of PRP4 in HCT116 and PC-3 cells. Specifically, we tested whether PRP4 phosphorylates p53 in these cells. The overexpression of PRP4 showed increased kinase activity in HCT116 and PC-3 cells (Appendix A). However, when we performed immunoprecipitation with PRP4 and p53 antibodies followed by kinase assay using the p53 as the substrate in the kinase assay, we noted that the PC-3 cells did not form adenosine diphosphate (ADP) (Appendix A). This result meant that p53 was not expressed in PC-3 cells, was not detected in immunoprecipitation with p53’s antibody, and did not form ADP in the kinase activity assay. We analyzed the expression of p21 using RT-PCR in control and PRP4-overexpressing HCT116 cells to link PRP4-mediated activation of p53 with its downstream target gene, p21 [38]. The overexpression of PRP4 resulted in downregulation of p21 transcript levels (Figure 1C and Appendix A), which implies that the p53/p21 signaling pathway might not be directly related to the activation of p53 by PRP4.

### 2.2. PRP4 Promotes Colorectal Cancer and Activates p53’s Expression In Vivo

Previously, we conducted in vitro studies and reported that PRP4 promoted drug resistance in cancer cell lines [31,39]. Here, we analyzed the role of PRP4 in vivo, using a subcutaneous xenotransplant tumor model by injecting human HCT116 cells into BALB/c-n mice. Figure 2A illustrates this protocol. Briefly, we injected parental HCT116 cells on the left side of mice and PRP4-overexpressed HCT116 cells on the right side of mice (Figure 2B). One-week postimplantation, we initiated the treatment with resveratrol in one group of mice (50 mg/kg i.p. injection), and the other group received vehicle injections. We measured the tumor sizes at 3–4-day intervals until 50 days postimplantation of cells. The PRP4-overexpressed HCT116 cell tumors grew more rapidly than the parental HCT116 cell tumors in the vehicle-treated group. There were no bodyweight differences between groups (Figure 2B). The tumors generated by PRP4-overexpressed HCT116 cells grew more rapidly in the resveratrol-treated mice group, but resveratrol effectively reduced the tumor size resulting from the implanted parental HCT116 cells (Figure 2B). We measured tumor volume according to the formula we described in Materials and Methods.

First, we performed Hematoxylin and Eosin (H&E) staining of the excised tumor tissue sections to explore how PRP4 promoted tumor growth in mice. The tumors generated by PRP4-overexpressed HCT116 cells included more erythrocytes than those generated by the parental HCT116 cells, suggesting that PRP4 might promote angiogenesis, a well-known process required by invasive tumors to grow and to metastasize [40]. The resveratrol treatment induced nuclear fragmentation, a typical marker for apoptotic cells. There was no nuclear fragmentation in both the PRP4-transfected or the implanted with parental control cells groups (Figure 2C).

With these results, we hypothesize that PRP4 may trigger the p53 gene’s activation, inducing EMT and drug resistance, ultimately causing increased tumor volume. Therefore, we determined the transcript and immunoreactive protein levels of p53 in the mouse tissues, along with EMT marker, and E-cadherin (the increased levels of PRP4 protein after PRP4 overexpression have not been shown intentionally in this set of data as well as in the next figures). The PRP4 activates p53 and inhibits E-cadherin (Figure 2D,E), suggesting that the specific PRP4–p53 protein interaction induces EMT and resistance to resveratrol and the eventual progression of colon cancer in vivo. Moreover, we utilized p53^−/−^ HCT116 cells, where PRP4 overexpression failed to drive the cells towards EMT phenotype (Figure 3A,B), which further confirmed that PRP4-mediated EMT was correlated with p53.

### 2.3. miRNA Profiling and Data Analysis in Parental and PRP4-Transfected Colon Cancer Cells

We analyzed the miRNA expression profiles of control and PRP4-transfected HCT116 cells with microarray analysis to identify differentially regulated miRNAs in response to PRP4 overexpression. The threshold we used for screening up- or downregulated miRNAs was fold-change ≥1.5 or ≤1.0, respectively. We identified 13 upregulated and 2 downregulated miRNAs in PRP4-transfected cells as compared with their parental counterparts (Figure 4A and Appendix A). We identified mir-142-3p, mir-146a-5p, mir-32-5p, mir-155-5p, mir-142-5p, mir-223-3p, mir-210, mir-96-5p, mir-194-5p, mir-376c-3p, mir-143-3p, mir-141-3p, and mir-21-5p among the upregulated miRNAs (Appendix A). Nonetheless, the levels of mir-18a-5p and mir-144-3p were downregulated (Appendix A). Therefore, the PRP4 overexpression might regulate the miRNA levels of HCT116 cells.

### 2.4. PRP4 Mediates Drug Resistance and Antiapoptotic Activity via miR-210’s Upregulation 

Among the PRP4-transfected HCT116 cells’ upregulated miRNAs, we focused on miR-210, previously identified as cell proliferation promotor [33,34,35]. We hypothesized that PRP4 might promote drug resistance and antiapoptotic activity by upregulating miR-210. We silenced miR-210 in HCT116 cells with si-miR210 and treated them with resveratrol to experimentally investigate this hypothesis. A flow cytometry analysis revealed that PRP4 blocked resveratrol-induced apoptosis in HCT116 cells (Figure 4B). The resveratrol-induced apoptosis in cells subjected to miR-210 silencing was almost restored (Figure 3B), suggesting that PRP4 might mediate antiapoptotic activity and drug resistance via upregulation of miR-210. Similar results were obtained with a dichlorofluorescein diacetate (DCFDA) assay and DAPI staining, indicating that miR-210 silencing inhibited the PRP4-mediated activity (Figure 4C,D). In earlier work, we reported that PRP4 activated cell survival pathways [31]. We performed Western blot analysis with total cell extracts from parental, PRP4-, and/or si-miR210-transfected cells to determine the relationship between PRP4-induced cell survival and upregulation of miR-210. Following transfection with a PRP4-expression plasmid in HCT116 cells, the protein level of Raf and p-Raf increased. PRP4 overexpression in HCT116 cells also increased ERK phosphorylation, and significantly upregulated c-MYC and HIF-1α expressions. miR-210 silencing significantly inhibited the survival of PRP4-induced cell (Figure 4E). The PRP4-induced impact on the morphology of HCT116 cells was also reversed (Figure 4F) in response to si-miR210 (i.e., a transition from round to flattened cells). These results suggest that PRP4 mediated drug resistance and antiapoptotic activity via upregulation of miR-210.

### 2.5. PRP4 Activates p53 via Upregulation of miR-210, Inducing EMT and Inhibiting the HCT116 Cell’s Invasion

We found that overexpression of PRP4 resulted in a profound reduction in the E-cadherin levels detected in HCT116 cells with immunofluorescence microscopy (Figure 5A). We hypothesized that miR-210 might mediate PRP4-induced E-cadherin loss. For this purpose, we investigated the invasion of HCT116 cells using the Boyden Millipore chamber system. HCT116 cells were seeded onto the upper chamber membrane of wells of a 24-well plate containing 500 µL cell culture medium. Cells were transfected with PRP4 plasmid and/or si-miR210. Afterwards, non-invading cells remaining on the membrane’s upper surface with aseptic cotton swabs at the specified times. The miR-210’s silencing results in E-cadherin levels’ restoration in the HCT116 cells, suggesting that miR-210 might impact the underlying PRP4-induced E-cadherin loss mechanism (Figure 5A). The miR-210 silencing confirms its involvement in the PRP4-induced inhibition of the HCT116 cell invasion (Figure 5B). From Figure 5B, one contradictory mechanism is highly surprising to us, i.e., on the one hand, PRP4 overexpression reduced the expression of E-cadherin, whereas, on the other hand, it blocked the invasion of HCT116 cells. These results are consistent with our previous report [31]. We could not find the facts behind this strange behavior of PRP4; however, we postulate that PRP4, through its potent mRNA splicing regulatory activity, alters the expression of EMT-associated genes in HCT116 cells to promote this transition. Moreover, it alters the expression of various other yet unknown proteins that are involved in increasing drug resistance and reducing cell invasion. Our Western blot and RT-PCT analyses revealed that the overexpression of PRP4 did not impact the total and phosphorylated p53 with miR-210 silencing (Figure 5C–E). This result suggests that PRP4 induces EMT and blocks the invasion of HCT116 cells via the upregulation of both miR-210 and p53.

### 2.6. The Action of PRP4 Is Mediated through the Activation of HIF-1α and miR-210 

Several studies have pointed to miR-210 as an essential target of HIF-1α [41,42,43]. As shown in Figure 4E, PRP4 positively regulates the expression of HIF-1α. We either treated the HCT116 cells with deferoxamine (DFO) or silenced HIF-1α expression and transfected the PRP4’s overexpression promotor vector to elucidate the relationships among overexpression of PRP4, activation of HIF-1α, and p53. The Western blot and the RT-PCR analyses revealed that PRP4 and DFO elevated the p53 levels, suggesting activation of p53 by PRP4 via positive regulation of HIF-1α. An unexpected finding is that the combined effect of PRP4 and DFO did not increase the p53 levels (Figure 6A,C (gel left side)). On the other hand, si-HIF reversed the PRP4-induced p53 expression and mRNA levels (Figure 6B,C (gel right side)). These results revealed that PRP4 transfection consecutively regulates HIF-1α, miR-210, and p53, potentially leading to EMT. Then, we used a flow cytometry assay to determine whether DFO impacted PRP4-induced drug resistance and antiapoptotic activity. The DFO had no appreciable proapoptotic or antiapoptotic impact on the HCT116 cells, but it enhanced PRP4-mediated antiapoptotic activity (Figure 6D). Differently, the si-HIF inhibited the PRP4-mediated antiapoptotic effects on the HCT116 cells (Figure 6D). The DFO also did not reverse the PRP4-mediated altered morphology of HCT116 cells, which was effectively restored by si-HIF (Figure 6E). These results support our hypothesis that PRP4 activates HIF-1α, miR-210, and p53, all considered critical factors in the underlying mechanism of PRP4-induced drug resistance.

### 2.7. Colon Tumor Induction by PRP4 Is Mediated through the Activation of HIF and miR-210 In Vivo

Next, we investigated whether or not the silencing of HIF-1α and miR-210 would prevent PRP4-induced xenotransplant tumors in nude mice. One group of mice received the parental HCT116 cells on their left side and the PRP4-transfected HCT116 cells on their right side. The second group received silenced (si-HIF/si-miR210-transfected) cells on their left side and the PRP4-overexpressing and silenced (si-HIF/si-miR210-transfected) cells on their right side. We measured tumor sizes at 3- or 4-day intervals until 50 days postimplantation of cells. Tumor growth was enhanced in the PRP4-transfected HCT116 cells as compared with the parental HCT116 cells. Still, we did not observe bodyweight differences. However, HCT116 cells transfected with si-HIF and si-miR210, with or without PRP4 overexpression before implantation, prevented tumor growth (Figure 7A,B). The tissue of the silenced group included fewer erythrocytes than those from the PRP4-overexpression group (Figure 7C). Finally, our immunofluorescence assays showed that si-miR210 stimulated p53 and inhibited E-cadherin (Figure 7D). These results suggest that activation of both HIF and miR-210 mediates PRP4-induced colon tumor growth.

## 3. Discussion and Conclusions

p53 is the most frequently inactivated gene in human cancers [44]. Moreover, mutant p53 protein loses WT p53 tumor suppression functions and concomitantly acquires recent oncogenic actions [45]. Studies with mouse models have shown that genetic reconstitution of WT p53 tumor suppression functions rescued tumor growth [46]. It is believed that either restoring WT p53 activity or blocking mutant p53 oncogenic activity could provide an efficient strategy to treat human cancers [45].

The appearances of WT p53 in HCT116 and HCT-15 cells are an engaging, ambiguous, and debatable phenomenon. Why does the p53 appear in these cells in normal form? Why is it not mutated? Why does this normal p53 not suppress cancer? What are the functions of p53 in cancer cells? These and other similar questions demand logical answers. Here, we noted that PRP4 transfection stimulated the expression of p53 in HCT116, HCT-15, and Colo 320 cells.

As compared with HCT116 and HCT-15 cells, the p53 protein did not appear in PC-3 cells, although it was visible in mRNA levels. It has been shown that p53 is mutant in PC-3 cells [37]. PRP4 transfection also failed to induce expression of p53 in PC-3 cells. Since certain proteasomes could degrade the p53 protein, we utilized potent proteasome inhibitors followed by a Western bot analysis, but we did not find p53 expression in PC-3 cells (Appendix A).

We observed that PRP4 associates with the p53 protein in coimmunoprecipitation assays and that knockdown of PRP4 with specific si-RNA resulted in decreased expression of p53. Currently, it is precisely not clear which of the PRP4’s protein regions directly interact with p53. It is fascinating that PRP4 is coimmunoprecipitated with WT p53 only (HCT116 and HCT-15 cells), whereas it does not coimmunoprecipitate with mutant p53 in PC-3, HT29, and Colo 320 cells. Possibly, the mutations in the p53 protein led to structural and/or conformational changes in PRP4’s binding site to p53. It is also possible that the specific structural and conformational changes in mutated p53 protein cause other competing binding proteins to occupy the PRP4’s binding site. However, a particular indirect relation between PRP4 and mutant p53 is likely because PPR4 regulated p53 expression in HT29 and Colo 320 cells. The PRP4’s kinase domain is undoubtedly a candidate for this interaction. The kinase activity assay also revealed that PRP4 exhibits kinase activity and phosphorylates p53 in HCT116 cells. Detailed structural studies of both PRP4 and p53 protein will help to elucidate this interaction. 

Many human cancers have mutations in p53 [47]. Mutant p53 forms have been shown to induce invasion and metastasis by regulating critical signaling pathways, including the integrin receptor and epidermal growth factor pathways that regulate the EMT via the ZEB1/2 transcriptional regulator [48,49]. In colon cancer cells, p53 with point mutations results in downregulated E-cadherin levels via a mechanism dependent on the actions of transcription factors Snail2 and ZEB1 [50]. Here, we observed PRP4 blocking resveratrol’s proapoptotic activity in vivo and promoting colon cancer PRP4 also generates an angiogenic phenotype in HCT116 cell-derived tumors. We propose that p53 induces EMT in the xenotransplant tumors after activation by PRP4. These factors ultimately lead to drug resistance and more extensive tumors. The overexpression of PRP4 did not reverse the RSV-induced cell death in the PC-3 cell lines (Appendix A) as it did in both the HCT116 and the HCT-15 cells. Moreover, PRP4 did not alter the morphology of PC-3 cells, supporting p53’s contribution to PRP4-induced drug resistance and EMT.

The current study describes the regulatory impact of PRP4 overexpression on miRNA expression levels. We conducted a microarray analysis to determine miRNA expression in PRP4-transfected cells. We focused on miR-210, since multiple studies have reported that it is consistently overexpressed in both normal and hypoxic cells [34,51,52,53,54,55]. Multiple miR-210’s targets were reported that cause numerous disease processes. Likewise, after activation via HIF, miR-210 induces angiogenesis [56]. Here, we silenced miR-210 with si-miR210 and conducted Western blot analyses and reverse transcription polymerase chain reaction (RT-PCR) that revealed that PRP4-induced activation of p53 was inhibited when miR-210 was silenced. Therefore, we conclude that PRP4 first activates the expression of miR-210, and then triggers the activation of p53. 

In this study, we used DFO and si-HIF and obtained experimental findings from Western blotting, flow cytometry, and immunofluorescence microscopy. We verified that PRP4-induced transcription of miR-210 and HIF-1α was directly linked to upregulation of p53. PRP4 enhances drug resistance, antiapoptosis, and remodeling of the cell cytoskeleton, while si-HIF abolishes PRP4 action in both HCT16 cell lines and xenotransplant tumors. Our study results, collectively suggest that PRP4-activated miR-210 is associated with p53 upregulation, and PRP4 may be a critical mediator of EMT and drug resistance. We conclude that PRP4 may be a potential target for anticancer therapies, as its specific inhibition could be of immediate benefit for patients with colon cancer and cancer in general.

## 4. Materials and Methods

### 4.1. Chemicals and Reagents

We purchased the cell lines from the American Type Culture Collection (Manassas, VA, USA). We obtained the Dulbecco’s modified Eagle’s medium (DMEM), RPMI-1640 medium, McCoy’s 5A medium, fetal bovine serum (FBS), and penicillin/streptomycin from Gibco (Carlsbad, CA, USA). We purchased the H&E Stain Kit (cat. no. H-3502) from Vector Laboratories, Burlingame, CA, USA. We obtained the Alexa Fluor 488 Phalloidin and 2′, 7′- DCFDA from Invitrogen (Eugene, OR, USA). We obtained the Annexin-V-FITC Apoptosis Detection Kit (ab14085) and Universal Kinase Assay Kit (ab138879) from Abcam (Cambridge CB2 0AX, UK). We obtained the ECM Invasion Chambers from Merck Millipore (Temecula, CA, USA). We purchased the PRP4 cDNA clone from Sino Biological (North Wales, PA, USA), and we obtained the PRP4 siRNA from Santa Cruz Biotechnology (SC-76257). We purchased resveratrol from Santa Cruz Biotechnology (CAS 501-36-0, Dallas, TX, USA). We obtained the LentimiRa-GFP-hsa-miR-210-3p vector (cat. no. mh10277) from Applied Biological Materials (Richmond, BC, Canada). We purchased the SuperScript III Reverse Transcriptase (cat. no. 18080093) and Lipofectamine LTX with Plus Reagent (cat. no. 15338100) from Invitrogen. We obtained the Xfect RNA transfection reagent (cat. no. 631450) from Takara Bio (Mountainview, CA, USA). We acquired the antibodies from Santa Cruz Biotechnology (β-actin (cat. no. sc-47778), ERK (cat. no. sc-292838), p-ERK (cat. no. sc-7383), Raf (cat. no. sc-133), p-Raf (cat. no. sc-28005), and c-Myc (cat. no. sc-40)) and Cell Signaling Technology (HIF-α (cat. no. 3716), p-p53 (cat. no. 9284), and E-cadherin (cat. no. 3195)). We purchased the proteasome Inhibitor Set II (cat. no. 539165) from Merck (USA). All the chemicals and reagents were used as directed by the manufacturers.

### 4.2. Cell Culture

We cultured the HCT116 (ATCC CCL-247), PC-3 (ATCC CRL-143), HCT-15 (ATCC CCL-225), Colo 320 (ATCC CCL-220), and HT-29 (ATCC HTB-38) cells in DMEM, RPMI-1640, or McCoy’s 5A medium supplemented with 10% FBS and 2% penicillin/streptomycin. We maintained the cultured cells at 37 °C in a humidified atmosphere containing 5% CO_2_.

### 4.3. Immunoprecipitation

We performed all procedures at 4 °C. We added the phenylmethylsulfonyl fluoride (PMSF) and the β-mercaptoethanol to cold solutions before initiating the experiment. After harvesting, we washed the cells with ice-cold phosphate-buffered saline (PBS), and then we resuspended them in 10 cell pellet volumes of TETN-150 buffer (0.5 mM PMSF, 0.5 mM EDTA, 0.5% Triton X-100, 20 mM Tris-HCl (pH 7.4), 2 mM MgCl_2_, 150 mM NaCl, and 5 mM β-mercaptoethanol). We incubated the cells at 30 min with rotation. We cleared the cells extracts by centrifugation at 18,000× *g* for 15 min and we transferred the supernatants to fresh tubes. We added the antibody and proceeded with incubation for 3 h, followed by a 45 min incubation with Protein A/G PLUS (Santa Cruz Biotechnology, cat. no. sc-2003). We eluted the immunoprecipitates by incubation in sodium dodecyl-sulfate (SDS), loading a buffer for 3 min at 95 °C. Then, we analyzed the samples by Western blot.

### 4.4. In Vitro Kinase Assay

We used the universal kinase assay kit (abcam) to monitor the in vitro kinase activity of PRP4. After, we transfected the HCT116 and PC-3 cells with PRP4, and we extracted their cDNA in ice-cold cell lysis buffer by sonication. We centrifuged the extracts at 10,000× *g* for 20 min at 4 °C. We immunoprecipitated the PRP4 and p53 (as a substrate for the kinase assays) proteins from the supernatant with the Protein A/G PLUS. We started the kinase reaction by adding the kinase reaction mixture (ADP assay buffer, 100 µM ATP, 10 mM MgCl_2_, 4 mM MnCl_2_, and Roche protease inhibitor cocktail), containing p53 protein as a substrate, to the purified immunoprecipitated PRP4 proteins and incubated them for 45 min at room temperature. We combined 20 µL of the kinase reaction mixture with 20 µL of ADP sensor buffer and 10 µL of ADP sensor to monitor the ADP formation, and we incubated the mixture in the dark for 1 h. We detected the fluorescence intensity of ADP products with a microplate fluorescence reader at Ex540 nm/Em590 nm.

### 4.5. miRNA and HIF-1α Silencing Using a pLenti-GFP Lentivirus Vector

We transfected the parental and PRP4-transfected cells with LentimiRa-GFP-hsa-miR-210-3p vector, according to the manufacturer’s protocol for the silencing of miR-210 in the HCT116 cells. We analyzed the impact of miR-210 silencing as described below. Additionally, HIF-1α was also silenced with lentiviral transfection. In this manuscript, miRNA-210- and HIF-1α-silenced cells are referred to as si-miR210 and si-HIF, respectively.

### 4.6. Annexin-V-FITC Apoptosis Assay

We performed the apoptosis assay as directed by the manufacturer (Abcam, cat. no. ab14085). Briefly, we cultured the cells in 6-well plates (2 × 10^5^ cells per well), transfected them with PRP4, and/or incubated them with resveratrol. Additionally, we transfected the cells with si-miR210 or si-HIF or incubated them with DFO for 24 h. Then, we gently removed the cells from the plates with trypsin, collected them by centrifugation, and resuspended them in 500 μL 1X binding buffer. Then, we stained the cells with annexin-V-FITC and propidium iodide (PI) (50 mg/mL, 5 μL per sample) at 25 °C for 10 min in the dark. Finally, we analyzed them by flow cytometry (FACSARIA III, BD Biosciences, San Jose, CA, USA).

### 4.7. F-Actin Staining

We used the Alexa Fluor 488 and Alexa Fluor 594 phalloidin for F-actin visualization. Briefly, after removal of the growth medium, we washed the cells twice with PBS and fixed them with 4% paraformaldehyde for 15 min at room temperature. Then, we permeabilized the cells with 0.2% Triton X-100 for 5 min and we washed them 2–3 times with PBS. We diluted the Alexa Fluor 488 and 594 phalloidin stock solutions (6.6 µM in methanol) 1:40 with 1% bovine serum albumin (BSA) and we added them to the cells for 50 min at room temperature in the dark. We washed the cells 5–6 times with PBS, and we observed the actin cytoskeleton using a ZEISS LSM 800 confocal microscope.

### 4.8. Immunofluorescence Microscopy

We fixed the cells we grew in a 4-well chamber slide with 4% paraformaldehyde for 30 min, followed by permeabilization with 0.2% Triton X-100. We blocked the cells with 2% BSA for 2–3 h, at 4 °C. We applied the anti-E-cadherin polyclonal antibody (1:500 in 2% BSA) overnight at 4 °C, followed by incubation with Alexa Fluor 594-labeled goat anti-rabbit secondary antibody (1∶100 with 2% BSA) for 1 h at room temperature in the dark. We washed the cells 5–6 times with PBS, and we counterstained the nuclei with DAPI mounting solution (cat. no. H-1200, Vector Laboratories, USA) for 5–10 min. We examined the cells with a ZEISS LSM 800 confocal microscope.

### 4.9. Cell Invasion Assay

We investigated the HCT116 cell invasion using the Boyden Millipore chamber system. Briefly, we incubated the inserts in serum-free medium for 2–3 h at room temperature. Then, we seeded the cells onto the upper chamber membrane of wells of a 24-well plate containing 500 µL DMEM. Cells were transfected with PRP4 plasmid and/or si-miR210. Next, we removed the non-invading cells remaining on the membrane’s upper surface with aseptic cotton swabs at the specified times. For visualization, we stained the HCT116 cells invading through the collagen to the membrane’s lower surface with 5% Giemsa solution for 30 min at room temperature in the dark. We washed the cells with PBS, air-dried, captured the images, and subjected them to quantitative evaluation using a Nikon SMZ18 system.

### 4.10. miRNA Microarray

According to the manufacturer’s instructions, we isolated the total RNA from parental and PRP4-transfected HCT116 cells using TRIzol reagent and miRNeasy Mini Kit (Qiagen, Hilden, Germany). For each sample, 1 µg of total RNA was 3′-end-labeled with the HY3TM fluorescent label using the miRCURY™ HY3™/Hy5™ Power Labeling kit (Exiqon, Vedbaek, Denmark) and hybridized to miRCURY™ LNA Arrays (version 18.0), according to the manufacturer’s instructions. The seventh generation of miRCURY™ LNA Arrays (Exiqon) contains 3100 capture probes covering all human, mouse, and rat miRNAs annotated in miRBase 18.0. In addition, this array includes capture probes for 25 miRPlus™ human miRNAs. We washed the slides several times after hybridization using the Wash buffer kit (Exiqon) and dried them by centrifugation for 5 min at 400 rpm. Then, we scanned the slides using the Axon GenePix 4000B microarray scanner (Axon Instruments, Foster City, CA, USA). We imported the scanned images into the GenePix Pro 6.0 software (Axon Instruments) for grid alignment and data extraction. We normalized the data to the median values, and we identified the differentially expressed miRNAs through fold change filtering. 

### 4.11. RT-PCR

For cDNA synthesis, we reverse-transcribed the total RNA (5 µg) using the SuperScript III First-strand synthesis kit, described previously [39]. We incubated the synthesized cDNA with RNase H at 37 °C for 2 h. We performed a PCR using 2 μL of cDNA and primers that include p21 forward, 5′-GTCCGTCAGAACCCATGC-3′ and p21 reverse, 5′-GTCGAAGTTCCATCGCTCA-3′; p53 forward, 5′-CCTCACCATCATCACACTGG-3′ and p53 reverse, 5′-CCTCATTCAGCTCTCGGAAC-3′; E-cadherin forward, 5′-TTGGCTCTGC CAGGAGCCGG-3′ and E-cadherin reverse, 5′-TGTCGACCGGTGCAATCTTC-3′; glyceraldehyde-3-phosphate dehydrogenase (GAPDH) forward, 5′-AGGGCTGCTTTTAACTCTGGT-3′ and GAPDH reverse, 5′-CCCCACTTGATTTTGGAGGGA-3′. We performed the PCR under the following conditions: one cycle at 98 °C for 3 min, followed by 30–35 cycles at 95 °C for 30 s, 55 °C for 30 s, and 72 °C for 30 s, with a final extension step at 72 °C for 5 min. We analyzed the amplified PCR products by 1.7% agarose gel electrophoresis and EcoDye Nucleic Acid Staining Solution (Biofact). Then, we captured the images using the Wise Capture I-1000 software (Daihan Scientific, Seoul, Korea).

### 4.12. Western Blot

We centrifuged the cells collected into PBS with a cell scraper at 12,000 rpm for 5 min to obtain the cell pellet. We discarded the supernatant and resuspended the cell pellet in 200 µL of cell lysis buffer (50 mM Tris, pH 7.4, 0.5% NP40, 0.01% SDS, and protease inhibitor cocktail (Roche, Penzberg, Germany)). According to the manufacturer’s protocol after lysis by sonication, we quantified total protein using the Bio-Rad Protein Assay. We prepared the samples (20–40 μg) in SDS sample buffer containing 60 mM Tris-HCl (pH 6.8), 2% SDS, 10% glycerol, and 5% β-mercaptoethanol, separated them by 10–12% SDS-polyacrylamide gel electrophoresis (PAGE), and transferred them onto a polyvinylidene fluoride membrane (Amersham, Piscataway, NJ, USA). We blocked the membranes with 3% albumin (genDEPOT, Barker, TX, USA) solution for 2 h, at 4 °C. We developed the chemiluminescent signals using a Clarity ECL Western Blotting Substrate (Bio-Rad, Hercules, CA 94547, USA), according to the manufacturer’s instructions.

### 4.13. Animal Study Protocol

We housed 20 male BALB/c-n mice at a density of five mice per cage, under conditions of constant temperature (22 °C) and a light/dark cycle of 12 h. The study was conducted according to the animal maintenance and use guidelines of Kyungpook National University (nos. KNU 2012-37 and 2016-42). We suspended a total of 10^6^ parental or PRP4-transfected and si-HIF/si-miR210-transfected HCT116 cells in 150 µL PBS and implanted them subcutaneously into 6-week-old mice at specific left and right sites using an insulin syringe. At one week postimplantation of cells, we divided the mice into two treatment groups, including (A) control (vehicle alone) and (B) resveratrol (50 mg/kg i.p. injection). We recorded tumor volumes weekly using a Vernier caliper and calculated them according to the formula: V = 4/3πW^2^L (short diameter^2^ × long diameter (mm^3^)). We excised the tumors 30–45 days postimplantation of cells using scissors. We formalin-fixed the excised tumors and paraffin-embedded them for subsequent histochemical staining.

### 4.14. H&E Staining

We applied hematoxylin to cover the tissue section completely, followed by incubation for 10 min. We rinsed the slides in two changes of distilled water (15 s each) to remove excess stain. Then, we applied the bluing reagent to cover the tissue section completely and incubated the slides for 10–15 s. Then, we rinsed the slides with two changes of distilled water. We dipped the slides in 100% ethanol for 10 s and applied Eosin Y Solution to cover the tissue section entirely for 2–3 min. We rinsed the slides with 100% ethanol for 10 s. We performed the dehydration with three changes of 100% ethanol (1–2 min each). We added the coverslips and examined the tissue sections under a light microscope.

### 4.15. Statistical Analysis

We prepared all samples in triplicate and repeated all experiments at least three times. We presented the data as mean ± standard deviation (SD). We evaluated the differences between groups using Student’s *t*-test, and *p*-values <0.05 were considered to be statistically significant.

## Figures and Tables

**Figure 1 ijms-23-03092-f001:**
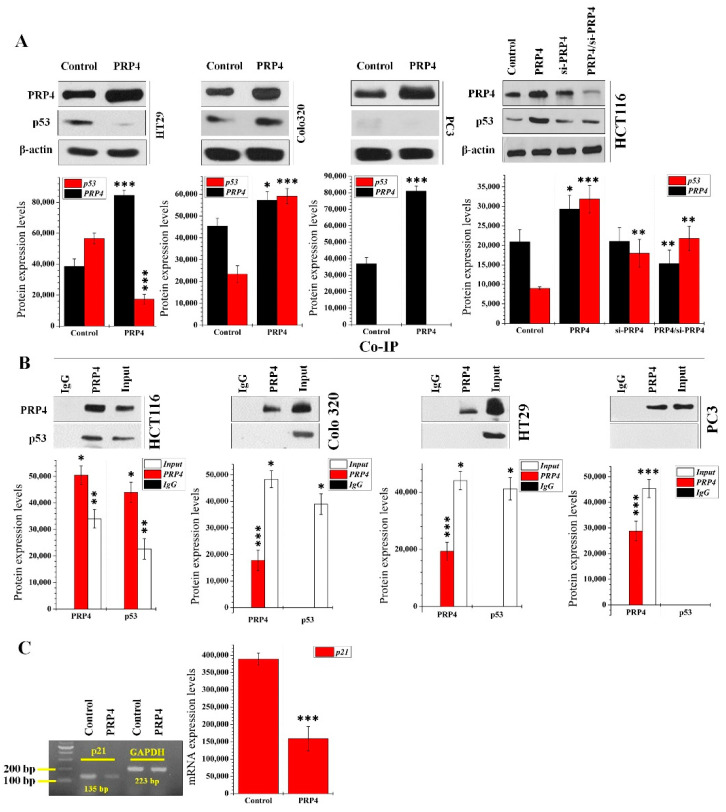
Interaction between PRP4 and p53: (**A**) Cell extracts from control and PRP4/si-PRP4 transfected groups were analyzed by Western blot with actin as a loading control. Data were collected from three independent experiments. * *p* < 0.05, ** *p* < 0.01, and *** *p* < 0.001. PRP4, transfection of PRP4 plasmid; si-PRP4, transfection of siRNA-PRP4; PRP4/si-PRP4, transfection with PRP4 plamsid and si-RNA-PRP4; (**B**) coimmunoprecipitation of p53 with an anti-PRP4 antibody. Samples of input (3% of total) and 30% of proteins from IP with antibody against PRP4, or control IgG from extracts of HCT116, Colo-320, HT29, and PC3 cells (1 mg) were separated by SDS-PAGE before reaction of Western blots with indicated antibodies. Data were collected from three independent experiments. * *p* < 0.05, ** *p* < 0.01, and *** *p* < 0.001; (**C**) expression of p21 at mRNA levels in parental and PRP4-transfected cells with glyceraldehyde-3-phosphate dehydrogenase (GAPDH) as an internal control. *** *p* < 0.001.

**Figure 2 ijms-23-03092-f002:**
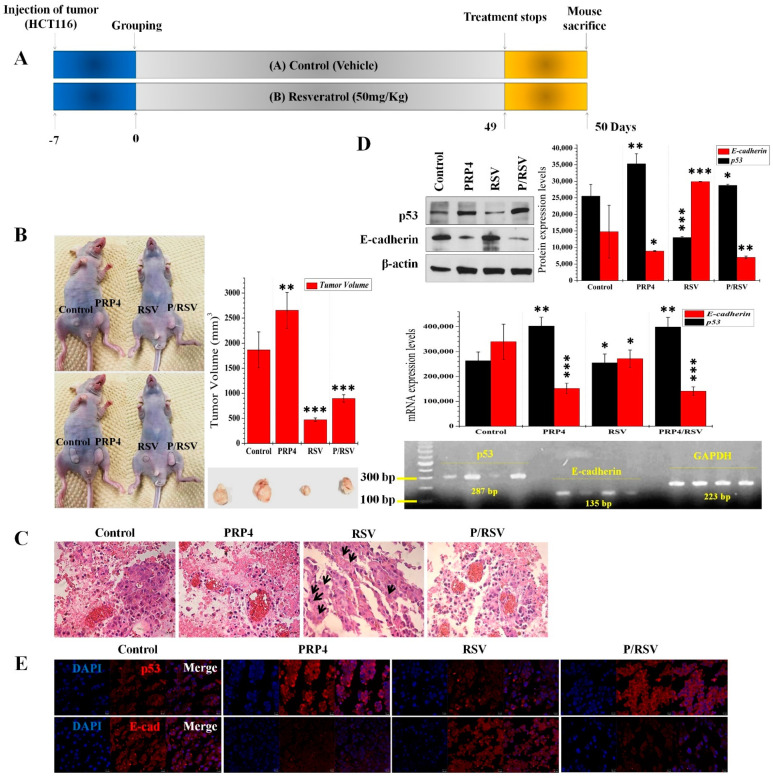
PRP4 promotes colon tumors in vivo: (**A**) Schematic of the in vivo experiment. BALB/c-n mice were treated with resveratrol (50 mg/kg body weight) or DMSO (vehicle) by intraperitoneal injection once daily, beginning one week and ending at six weeks after implantation of tumor cells, and mice were euthanized at Day 50; (**B**) the antitumor effects of resveratrol are blocked by PRP4 overexpression. Representative images of mice and tumors from each treatment group are as shown. BALB/c-n mice received parental HCT116 cells on the left side and PRP4-overexpressed HCT116 cells on the right side (left and right sides are according to the photo) by subcutaneous injection. Tumor volumes from individual mice are presented as mean ± standard deviation (*n* = 5 per group, Student’s *t*-test, ** *p* < 0.01, *** *p* < 0.001). For clear visibility of the tumors in vivo, the same mouse was photographed twice; the tumors are encircled in the latter set of photographs. RSV, treatment with resveratrol; P/RSV, transfection with PRP4 followed by treatment with 10 µM resveratrol; (**C**) sections of xenotransplant tumors from control and resveratrol-treated and/or PRP4-overexpressed groups were subjected to Hematoxylin and Eosin (H&E) staining. Cell staining includes light pink cytoplasm, red erythrocytes, pink collagen; blue nuclei, and pink/rose muscle tissue. Black arrowheads in the resveratrol-treated group indicate fragmented nuclei; (**D**) mouse tumor tissue extracts from control and treated groups were analyzed for expression of p53 and E-cadherin by Western blot and RT-PCR with actin and GAPDH as a loading control, respectively. * *p* < 0.05, ** *p* < 0.01, *** *p* < 0.001; (**E**) immunofluorescence microscopy of mouse tumor sections documenting the expression of p53 and E-cadherin. Tissues sections are immunostained with anti-p53 and anti-E-cadherin antibodies. Nuclei are visualized by DAPI staining.

**Figure 3 ijms-23-03092-f003:**
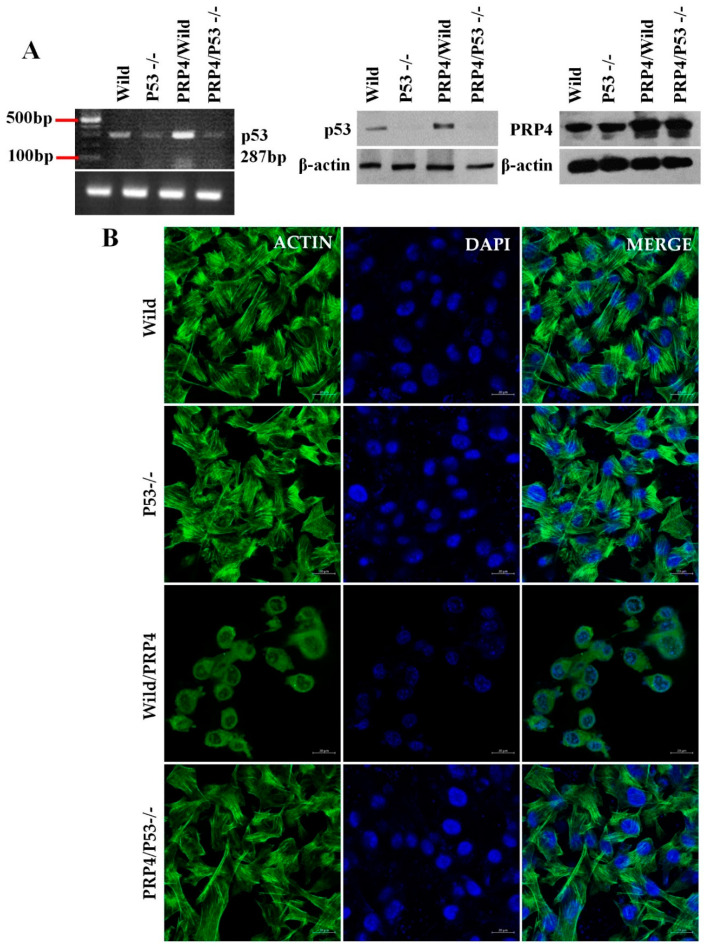
PRP4 overexpression fails to induce EMT phenotype in p53^−/−^ HCT116 cells: (**A**) Expression of p53 and PRP4 at mRNA and protein levels in HCT116 cells. Wild, HCT116 cells with wild-type p53; p53^−/−^, p53 deleted HCT116 cells; PRP4/Wild, HCT116 cells with wild-type p53 transfected with PRP4; PRP4/p53^−/−^, p53 deleted HCT116 cells transfected with PRP4; (**B**) HCT116 cells were stained with phalloidin and DAPI and observed under a confocal laser microscope at 400× magnification.

**Figure 4 ijms-23-03092-f004:**
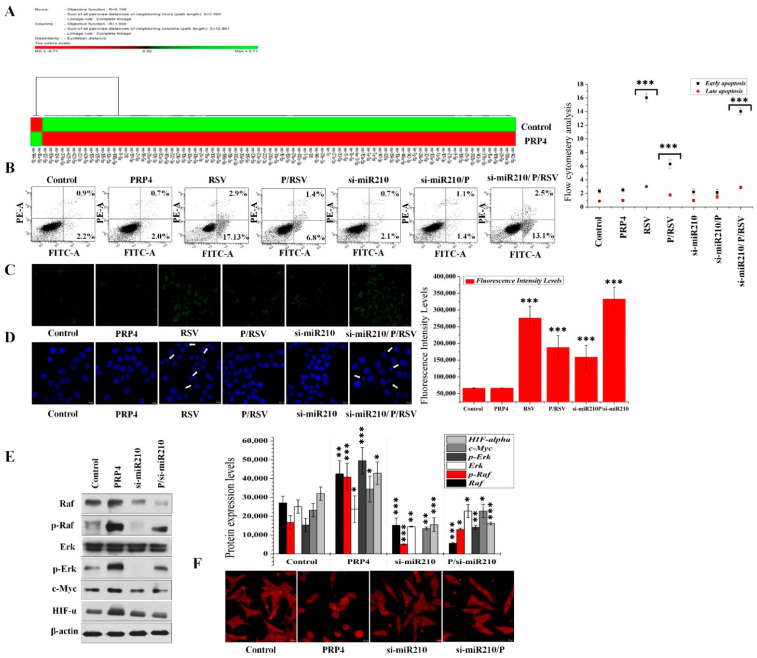
PRP4 induces drug resistance and modulates cell morphology by activating miR-210: (**A**) The green symbols represent miRNAs expressed at levels above the mean, the red symbols represent miRNAs expressed at levels below the mean; (**B**) the inhibitory impact of si-miR210 on PRP4 activity, lower and upper right panels represent early and late apoptotic cells, respectively. si-miR210, cells transfected with LentimiRa-GFP-hsa-miR-210-3p vector; si-miR210/P, cells transfected with si-miR210 and PRP4; si-miR210/P/RSV, cells transfected with si-miR210 and PRP4 followed by treatment with resveratrol. *** *p* < 0.001; (**C**) HCT116 cells transfected with a PRP4 expression plasmid or si-miR210 and incubated with or without 10 μM resveratrol (RSV) for 24 h. The cells were incubated with DCFHDA dye for 20 min to measure reactive oxygen species (ROS). Probe accumulation was measured in triplicate based on increased emission at 530 nm. ROS levels are expressed as the ratio of sample’s mean intensity to the control cells’ mean intensity. *** *p* < 0.001; (**D**) DAPI staining of cell nuclei, white arrowheads indicate fragmented nuclei; (**E**) Western blot analysis using antibodies (1:1000 dilution) specific for Raf, p-Raf, Erk, p-Erk, and c-Myc. HIF-1α with actin was used as an internal control. * *p* < 0.05, ** *p* < 0.01, *** *p* < 0.001; (**F**) F-actin phalloidin staining in HCT116 cells transfected with a PRF4-expression plasmid and/or si-miR210.

**Figure 5 ijms-23-03092-f005:**
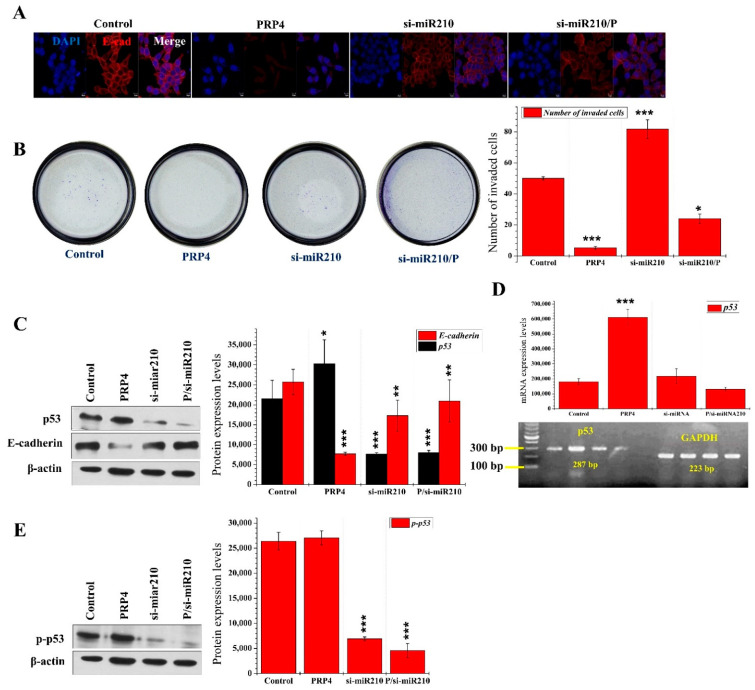
PRP4 induces epithelial–mesenchymal transition (EMT) and blocks cell invasion by upregulating miR-210 and p53: (**A**) Immunofluorescence microscopy images documenting E-cadherin expression in response to PRP4 overexpression and/or miR-210 silencing in HCT116 cells. Cells were immunostained with an anti-E-cadherin antibody, and nuclei are visualized by DAPI staining; (**B**) cellular invasion assay. Blue spots represent invading HCT116 cells. PRP4 overexpression results in a decreased invasion of HCT116 cells, while miR-210 silencing blocks PRP4 activity. * *p* < 0.05, *** *p* < 0.001; (**C**,**E**) Western blots were performed using antibodies (1:1000 dilution) specific for total and phosphorylated p53; actin was used as an internal control. * *p* < 0.05, ** *p* < 0.01, *** *p* < 0.001; (**D**) expression of p53 mRNA, GAPDH was used as a loading control. *** *p* < 0.001.

**Figure 6 ijms-23-03092-f006:**
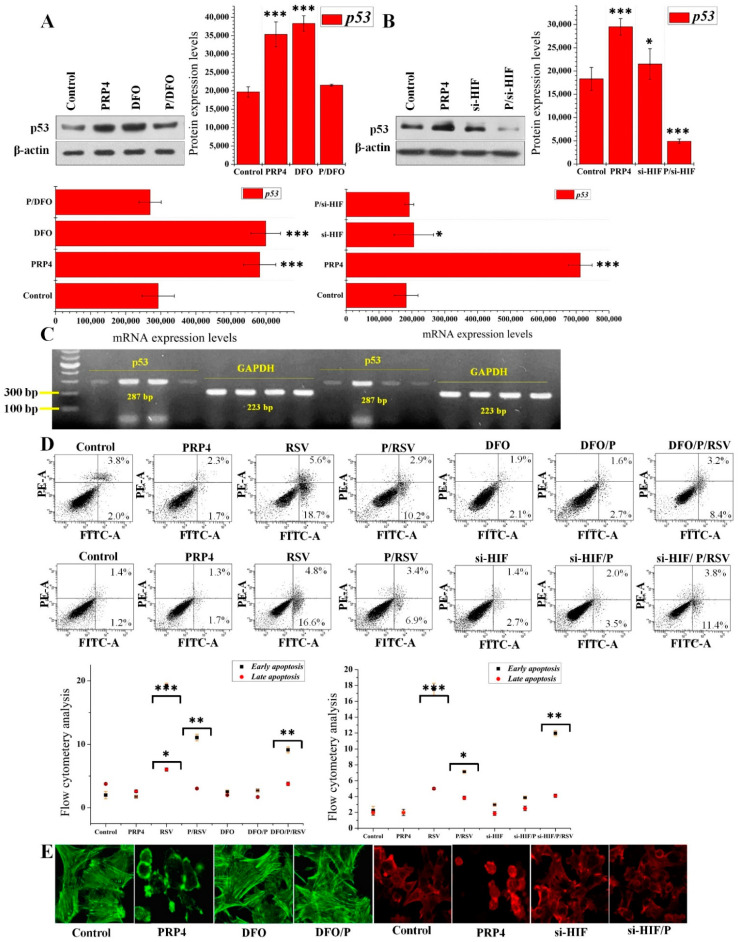
PRP4 action is mediated through the activation of HIF-1α and miR-210. Protein and mRNA levels of p53 in untransfected, PRP4-transfected, HIF-1α-mimetic-induced, and HIF-1α-silenced HCT116 cells were analyzed by (**A**,**B**) Western blot and (**C**) RT-PCR, with actin and GAPDH used as loading controls for Western blots and RT-PCR, respectively. * *p* < 0.05, *** *p* < 0.001. DFO, treatment with deferoxamine; P/DFO, treatment with deferoxamine and transfection with PRP4; si-HIF, transfection of siRNA-HIF 1α; P/si-HIF, transfection with PRP4 and siRNA-HIF 1α; DFO/P/RSV, treatment with deferoxamine and resveratrol and transfection with PRP4; si-HIF/P/RSV, transfection with si-RNA HIF 1α and si-RNA PRP4, and treatment with resveratrol; (**D**) the upper panel represents an annexin-V/PI apoptosis assay documenting the regulatory effect of 300 µM DFO on PRP4 activity, the lower and upper right panels represent early and late apoptotic cells, respectively, whereas, the lower panel represents an annexin-V/PI apoptosis assay documenting the regulatory effect of si-HIF on PRP4 activity, lower and upper right quarters represent early and late apoptotic cells, respectively. HIF-1α silencing effectively blocks PRP4-mediated antiapoptotic effects. * *p* < 0.05, ** *p* < 0.01, *** *p* < 0.001; (**E**) HCT116 cells were treated with DFO or transfected with si-HIF, and then transfected with a PRP4-expression plasmid. The cells were stained with phalloidin and observed under a confocal laser microscope at 400× magnification.

**Figure 7 ijms-23-03092-f007:**
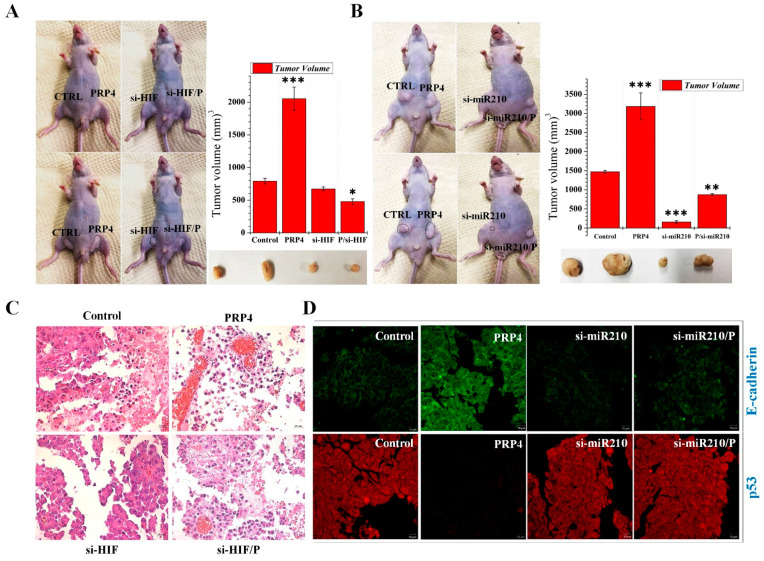
PRP4-induced tumors are associated with the upregulation of HIF-1α and miR-210 in vivo: (**A**,**B**) Representative images of xenotransplant tumors in nude mice. The first group were implanted with parental HCT116 cells (control, on the left as per the photograph) and PRP4-overexpressed HCT116 cells on the right. The second group received silenced (si-HIF/si-miR210)-transfected HCT116 cells on the left and PRP4-overexpresed and silenced (si-HIF/si-miR210-transfected) cells on the right. Tumor sizes were measured at 3- or 4-day intervals until 50 days after postimplantation of cells. For clear visibility of the tumors in vivo, the same mouse was photographed twice, and the tumors are encircled in the second photograph. Data shown are from three independent experiments. * *p* < 0.05, ** *p* < 0.01, *** *p* < 0.001; (**C**) sections of the xenotransplant tumors isolated from each of the four conditions were subjected to Hematoxylin and Eosin (H&E) staining. Cell staining includes light pink cytoplasm, red erythrocytes, pink collagen, blue nuclei, and pink/rose muscle tissue. The tumors subjected to HIF-1α silencing contained fewer erythrocytes than their counterpart PRP4-overexpressing tumors; (**D**) immunofluorescence microscopy of mouse tumor sections immunostained with anti-p53 and anti-E-cadherin antibodies.

## Data Availability

The data that support the findings of this study are available on request from the corresponding author.

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
