# Peer review of "PRP4 Induces Epithelial–Mesenchymal Transition and Drug Resistance in Colon Cancer Cells via Activation of p53"

_ijms, 2022, doi:10.3390/ijms23063092_

Round 1
Reviewer 1 Report
Dear Authors,
I have received your manuscript entitled ” PRP4 Induces Epithelial–Mesenchymal Transition and Drug Resistance in Colon Cancer Cells Via Activation of p53 ” to be considered for publication in International Journal of Molecular Sciences as an article. The work is focused on studying the interaction of PRP4 with p53 in various colon cancer cell lines and in vivo in mice to prove that inhibition of PRP4 may be a potential target of anti-cancer therapies. For this purpose, many molecular biology methods were used to assess the necessary molecules at the gene and protein level (RT-PCR, western blotting, flow cytometry and immunofluorescence microscopy). In addition, it was tested with miR-210 to see how PRP4 acts on p53. The work is based on many interesting results and at the same time they all confirm the Authors' hypothesis.
My little comment:
The paper presents the result of the microarray analysis in which there are many miRNAs and the only miR-210 was selected and used on the basis of the publication [34, 51-55] in Figure 4 (A) and in the supplement. Why are shown another significant miRNAs for PRP4?
Sincerely,
Reviewer
Author Response
Reviewer 1
I have received your manuscript entitled” PRP4 Induces Epithelial–Mesenchymal Transition and Drug Resistance in Colon Cancer Cells Via Activation of p53 ” to be considered for publication in International Journal of Molecular Sciences as an article. The work is focused on studying the interaction of PRP4 with p53 in various colon cancer cell lines and in vivo in mice to prove that inhibition of PRP4 may be a potential target of anti-cancer therapies. For this purpose, many molecular biology methods were used to assess the necessary molecules at the gene and protein level (RT-PCR, western blotting, flow cytometry and immunofluorescence microscopy). In addition, it was tested with miR-210 to see how PRP4 acts on p53. The work is based on many interesting results and at the same time they all confirm the Authors' hypothesis.
My little comment:
The paper presents the result of the microarray analysis in which there are many miRNAs and the only miR-210 was selected and used on the basis of the publication [34, 51-55] in Figure 4 (A) and in the supplement. Why are shown another significant miRNAs for PRP4?
Response: Thank you very much for your comment. We are interested in cell signaling which are related to cancer cells proliferation, growth, invasion, metastasis, progression, and anticancer drug resistance. miR-210 has been previously identified as cell proliferation promotor [Ref# 33-35 of the current manuscript], while all other PRP4-regulated miRNAs were related to other cellular processes. Therefore, we focused only miR-210 and extended our study analyzing its roles in induction of anticancer drug resistance and cancer progression correlated with PRP4. All other PRP4-regulated miRNAs will help us to explore multiple unknown mechanism of PRP4 in various cellular processes.

Reviewer 2 Report
The manuscript entitled “PRP4 Induces Epithelial–Mesenchymal Transition and Drug Resistance in Colon Cancer Cells Via Activation of p53” describes that PRP4-mediated miR-210’s activation acts on p53, a potentially-critical EMT and drug resistance mediator. This well-designed and well-written work may be of interest to IJMS readers. However, some minor issues should be addressed.
- TP53 is the gene that encodes p53 protein. Genes should be italicized and revised throughout the entire manuscript the use of p53 when talking about the TP53 gene.
- Results from HCT-15 cell line are not shown in Figure 1, and this fact should be mentioned in the text (line 92).
- Explain why resveratrol was used in the in vivo studies since this is not the standard treatment for colon cancer.
- In Figures 4B and 6D, the axes of flow cytometry dot plots are too small. Please improve these axes, including Annexin V and PI, to improve the graphics readability.
- On line 212, Fig. 3b should be replaced by Fig. 4B. On line 264, Fig. 4E should be replaced by Fig. 6E.
- In the Statistical analysis sub-section, the Authors described the use of the student’s t-test to compare groups. Did the Authors analyze the data normality? Were the p-values corrected for multiple comparisons?
Author Response
Reviewer 2
The manuscript entitled “PRP4 Induces Epithelial–Mesenchymal Transition and Drug Resistance in Colon Cancer Cells Via Activation of p53” describes that PRP4-mediated miR-210’s activation acts on p53, a potentially-critical EMT and drug resistance mediator. This well-designed and well-written work may be of interest to IJMS readers. However, some minor issues should be addressed.
TP53 is the gene that encodes p53 protein. Genes should be italicized and revised throughout the entire manuscript the use of p53 when talking about the TP53 gene.
Response: We highly appreciate the comment of the reviewer. p53 gene has been italicized throughout the manuscript.
Results from HCT-15 cell line are not shown in Figure 1, and this fact should be mentioned in the text (line 92).
Response: Thank you very much for your comment. The data of HCT-15 cells has been shown in Supplemental figure S1A, S1C, and S1E. It is added to line 92 as directed by the reviewer.
Explain why resveratrol was used in the in vivo studies since this is not the standard treatment for colon cancer.
Response: Thank you very much for your comment. We were interested to analyze the anticancer effects of naturally occurring phytochemicals like decursin angelate, curcumin, and resveratrol. Additionally, the current investigation is the expansion of our previous studies where we investigated the effect of resveratrol against HCT-15 and HCT116 cells.
In Figures 4B and 6D, the axes of flow cytometry dot plots are too small. Please improve these axes, including Annexin V and PI, to improve the graphics readability.
Response: Thank you very much for your comment. Although, these axes have been automatically inserted by the flow cytometer software; however, we are manually maximizing their size according to the reviewer’s suggestion. Please refer to the concerned figures.
On line 212, Fig. 3b should be replaced by Fig. 4B. On line 264, Fig. 4E should be replaced by Fig. 6E.
Response: Thank you very much for your suggestion. In line 212, correction has been made (3B>>4B). However, in line 264, 4E has been retained, as in this line we have referred to the western blot data shown in Figure 4E.
In the Statistical analysis sub-section, the Authors described the use of the student’s t-test to compare groups. Did the Authors analyze the data normality? Were the p-values corrected for multiple comparisons?
Response: Thank you very much for your comment. Yes, of course, we repeated all experiments at least three times, presented the data as mean ± standard deviation (SD), and evaluated the differences between groups with the student’s t-test. p-values <0.05 were considered statistically significant.
